# Low-frequency variation near common germline susceptibility loci are associated with risk of Ewing sarcoma

Shu-Hong Lin[1], Joshua N. Sampson[1], Thomas G. P. Grünewald[2,3,4], Didier Surdez[5], Stephanie Reynaud[6], Olivier Mirabeau[5,6], Eric Karlins[1,7], Rebeca Alba Rubio[2], Sakina Zaidi[5,6], Sandrine Grossetête-Lalami[5,6], Stelly Ballet[6], Eve Lapouble[6], Valérie Laurence[6], Jean Michon[6], Gaelle Pierron[6], Heinrich Kovar[8], Udo Kontny[9], Anna González-Neira[10], Javier Alonso[11], Ana Patino-Garcia[12], Nadège Corradini[13], Perrine Marec Bérard[13], Jeremy Miller[14], Neal D. Freedman[1], Nathaniel Rothman[1], Brian D. Carter[15], Casey L. Dagnall[1,7], Laurie Burdett[1,7], Kristine Jones[1,7], Michelle Manning[1,7], Kathleen Wyatt[1,7], Weiyin Zhou[1,7], Meredith Yeager[1,7], David G. Cox[16], Robert N. Hoover[1], Javed Khan[17], Gregory T. Armstrong[18], Wendy M. Leisenring[19], Smita Bhatia[20], Leslie L. Robison[18], Andreas E. Kulozik[21], Jennifer Kriebel[22,23,24], Thomas Meitinger[25,26], Markus Metzler[27], Manuela Krumbholz[27], Wolfgang Hartmann[28], Konstantin Strauch[29], Thomas Kirchner[30], Uta Dirksen[31,32], Lisa Mirabello[1], Margaret A. Tucker[1], Franck Tirode[5,6], Lindsay M. Morton[1], Stephen J. Chanock[1], Olivier Delattre[5,6], Mitchell J. Machiela[1]*

**Data Availability Statement:** EWS GWAS is available on dbGaP under accession number phs001549.v1.p1 Data from CCSS is available on dbGaP under accession number phs001327.v1.p1.

1 Division of Cancer Epidemiology and Genetics, National Cancer Institute, Bethesda, MD, United States of America, 2 Max-Eder Research Group for Pediatric Sarcoma Biology, Ludwig Maximilians Universität (LMU), Munich, Germany, 3 Division of Translational Pediatric Sarcoma Research, German Cancer Research Center (DKFZ), German Cancer Consortium (DKTK), Heidelberg, Germany, 4 Institute of Pathology, Heidelberg University Hospital, Heidelberg, Germany, 5 Inserm U830, Équipe Labellisés LNCC, PSL Université, Institut Curie, Paris, France, 6 SIREDO Oncology Centre, Institut Curie, Paris, France, 7 Cancer Genomics Research Laboratory, Frederick National Laboratory for Cancer Research, Leidos Biomedical Research Inc., Frederick, MD, United States of America, 8 Children's Cancer Research Institute, St. Anna Kinderkrebsforschung, Vienna, Austria, 9 Division of Pediatric Hematology, Oncology and Stem Cell Transplantation, Uniklinik RWTH Aachen, Aachen, Germany, 10 Human Genotyping Unit-CeGen, Human Cancer Genetics Programme, Spanish National Cancer Research Centre, Madrid, Spain, 11 Unidad de Tumores Solidos Infantiles (IIER-ISCIII) & Centro de Investigación Biomédica en Red de Enfermedades Raras (CB06/07/1009; CIBERER-ISCIII), Instituto de Salud Carlos III, Majadahonda, Spain, 12 Laboratory of Pediatrics, University Clinic of Navarra, Program in Solid Tumors, Center for Applied Medical Research (CIMA) and Navarra's Health Research Institute (IdiSNA), Pamplona, Spain, 13 Institute for Paediatric Haematology and Oncology, Leon Bérard Cancer Centre, University of Lyon, Lyon, France, 14 Information Management Services, Inc., Calverton, MD, United States of America, 15 Behavioral and Epidemiology Research Group, American Cancer Society, Atlanta, GA, United States of America, 16 Institut National de la Santé et de la Recherche Médicale (INSERM), Paris, France, 17 Genetics Branch, Center for Cancer Research, National Cancer Institute, Bethesda, MD, United States of America, 18 Department of Epidemiology and Cancer Control, St. Jude Children's Research Hospital, Memphis, TN, United States of America, 19 Cancer Prevention and Clinical Statistics Programs, Fred Hutchinson Cancer Research Center, Seattle, WA, United States of America, 20 Institute for Cancer Outcomes and Survivorship, University of Alabama at Birmingham, Birmingham, AL, United States of America, 21 Department of Pediatric Oncology, Hematology and Immunology and Hopp Children Cancer Center, University of Heidelberg, Heidelberg, Germany, 22 Research Unit of Molecular Epidemiology, Helmholtz Zentrum München, German Research Center for Environmental Health, Neuherberg, Germany, 23 Institute of Epidemiology, Helmholtz Zentrum München, German Research Center for Environmental Health, Neuherberg, Germany, 24 German Center for Diabetes Research (DZD), München, Neuherberg, Germany, 25 German Research Center for Environmental Health, Institute of Human Genetics, Helmholtz Zentrum München, Neuherberg, Germany, 26 Institute of Human Genetics, Technische Universität München, Munich, Germany, 27 Department of Paediatrics and Adolescent Medicine, University Hospital of Erlangen, Erlangen, Germany, 28 Division of Translational Pathology, Gerhard-Domagk Institute of Pathology, University Hospital of Münster, Münster, Germany, 29 Institute of Genetic Epidemiology, LMU Munich, Munich, Germany, 30 Institute of Pathology, Faculty of Medicine, LMU Munich, Munich, Germany, 31 Pediatrics III, West German Cancer Centre, University Hospital Essen, Essen, Germany, 32 German Cancer Consortium (DKTK), Center Essen,

**Funding:** This work was supported by the National Cancer Institute (CA55727, G.T. Armstrong, Principal Investigator), with additional funding for genotyping from the Intramural Research Program of the National Institutes of Health, National Cancer Institute and the Intramural Research Program of the American Cancer Society. This work was supported by grants from the Institut Curie, the Inserm, the Ligue Nationale Contre le Cancer (Equipe labellisée, Carte d'Identité des Tumeurs program and Recherche Epidémiologique 2009 program), the ANR-10-EQPX-03 from the Agence Nationale de la Recherche, the European PROVABES (ERA-649 NET TRANSCAN JTC-2011), and ASSET (FP7-HEALTH-2010-259348) projects. This research was supported by FP7 grant "EURO EWING Consortium" No. 602856 and the following associations: Courir pour Mathieu, Dans les pas du Géant, Les Bagouzamanon, Enfants et Santé, M la vie avec Lisa, Lulu et les petites bouilles de lune, les Amis de Claire, l'Etoile de Martin and the Société Française de lutte contre les Cancers et les leucémies de l'Enfant et de l'adolescent. The laboratory of T. G. P. Grünewald is supported by grants from the 'Verein zur Förderung von Wissenschaft und Forschung an der Medizinischen Fakultät der LMU München (WiFoMed)', by LMU Munich's Institutional Strategy LMU excellent within the framework of the German Excellence Initiative, the 'Mehr LEBEN für krebskranke Kinder —Bettina-Bräu-Stiftung', the Wilhelm Sander-Foundation (2016.167.1), the Barbara and Hubertus Trettner foundation, the Gert and Susanna Mayer foundation, the Matthias-Lackas foundation, the Friedrich-Baur foundation, the Dr. Leopold and Carmen Ellinger foundation, the Dr. Rolf M. Schwiete foundation, the Deutsche Forschungsgemeinschaft (DFG 391665916), the Barbara and Wilfried Mohr foundation, the SMARCB1 e.V. assoication, and by the German Cancer Aid (DKH-70112257). D. Surdez is supported by SiRIC (Grant « INCa-DGOS-4654). The Metzler lab received grants from the European Commission Seventh Framework Program FP7-HEALTH "Euro Ewing Consortium EEC", project number EU-FP7 602856, the "Schornsteinfeger helfen krebskranken Kindern" Foundation and the Trettner Foundation. The group of U. Dirksen is supported by the German Cancer Aid grant 108128, the Barbara and Hubertus Trettner foundation, the Gert and Susanna Mayer foundation; ERA-Net-TRANSCAN consortium ´PROVABES´ (01KT1310), and Euro Ewing Consortium EEC, project number EU-FP7 602856, both funded under the European Commission Seventh Framework Program FP7-HEALTH (http://cordis.europa.eu/); This work was supported by

Heidelberg, Germany

* mitchell.machiela@nih.gov

# Abstract

## Background

Ewing sarcoma (EwS) is a rare, aggressive solid tumor of childhood, adolescence and young adulthood associated with pathognomonic EWSR1-ETS fusion oncoproteins altering transcriptional regulation. Genome-wide association studies (GWAS) have identified 6 common germline susceptibility loci but have not investigated low-frequency inherited variants with minor allele frequencies below 5% due to limited genotyped cases of this rare tumor.

## Methods

We investigated the contribution of rare and low-frequency variation to EwS susceptibility in the largest EwS genome-wide association study to date (733 EwS cases and 1,346 unaffected controls of European ancestry).

## Results

We identified two low-frequency variants, rs112837127 and rs2296730, on chromosome 20 that were associated with EwS risk (OR = 0.186 and 2.038, respectively; P-value < $5 \times 10^{-8}$) and located near previously reported common susceptibility loci. After adjusting for the most associated common variant at the locus, only rs112837127 remained a statistically significant independent signal (OR = 0.200, P-value = $5.84 \times 10^{-8}$).

## Conclusions

These findings suggest rare variation residing on common haplotypes are important contributors to EwS risk.

## Impact

Motivate future targeted sequencing studies for a comprehensive evaluation of low-frequency and rare variation around common EwS susceptibility loci.

# Background

Ewing sarcoma (EwS) is a rare bone or soft tissue tumor predominantly occurring in the second decade of life [1]. The specific cells of origin leading to EwS tumors are unknown, with current evidence indicating EwS likely arises from mesoderm- or neural crest-derived mesenchymal stem cells [2,3]. The overall age-adjusted incidence of EwS is 0.128 per 100,000 population with individuals of European ancestry at a 9-fold risk relative to African Americans and Asian/Pacific Islanders (0.155 in White, 0.017 in Asians/Pacific islanders, and 0.017 in African Americans) [4]. The reported disparity in EwS incidence by ancestry suggests the importance of germline susceptibility to EwS risk.

the Instituto de Salud Carlos III (PI16CIII/00026) and the Asociación Pablo Ugarte, Fundación Sonrisa de Alex, ASION-La Hucha de Tomás, Sociedad Española de Hematología y Oncología Pediátricas. Support to St. Jude Children's Research Hospital also provided by the Cancer Center Support (CORE) grant (CA21765, C. Roberts, Principal Investigator) and the American Lebanese-Syrian Associated Charities (ALSAC). The KORA study was initiated and financed by the Helmholtz Zentrum München—German Research Center for Environmental Health, which is funded by the German Federal Ministry of Education and Research (BMBF) and by the State of Bavaria. Furthermore, KORA research was supported within the Munich Center of Health Sciences (MC-Health), Ludwig-Maximilians-Universität, as part of LMUinnovativ. The laboratory of A.P. Garcia is supported by Gobierno de Navarra, Proyectos de Biomedicina 2018. Ref. 54/2018 and Fundación Caja Navarra/La Caixa to Niños Contra el Cáncer. Leidos Biomedical Research Inc. and Information Management Services, Inc. provided support in the form of salaries for authors J.M., E.K., C.L.D., L.B., K.J., M.M., K.W., W.Z., and M.Y., but did not have any additional role in the study design, data collection and analysis, decision to publish, or preparation of the manuscript. The specific roles of these authors are articulated in the 'author contributions' section. The funders had no role in study design, data collection and analysis, decision to publish, or preparation of the manuscript.

**Competing interests:** The authors have read the journal's policy and the authors of this manuscript have the following competing interests: Leidos Biomedical Research Inc. and Information Management Services, Inc. provided salaries for authors J.M., E.K., C.L.D., L.B., K.J., M.M., K.W., and W.Z. This does not alter our adherence to PLOS ONE policies on sharing data and materials. There are no patents, products in development or marketed products to declare.

A defining feature of EwS tumors is the somatically acquired translocation between *EWSR1* (22q12) and a member of the *ETS* transcription factor family, most commonly *FLI1* (11q24) (85% of cases) [5–7]. The resulting fusion oncoprotein produces aberrant and strong transcriptional regulators that bind to GGAA microsatellites and ETS-like motifs, which are thereby converted into potent enhancers, to promote cellular transformation by deregulating key target genes in cell cycle control, migration and apoptosis pathways [7–12]. Aside from recurrent *EWSR1-ETS* fusions, most EwS tumors display remarkably low somatic mutation rates [1,13–16].

The presence of EwS *EWSR1-ETS* fusions provides a molecularly distinct phenotype for genomic characterization, despite small case sample sizes. Previous genome-wide association studies (GWAS) have identified 6 common genetic susceptibility loci associated with EwS risk (1p36.22, 6p25.1, 10q21, 15q15, 20p11.22 and 20p11.23) [17]. The number of identified susceptibility loci are notable given small samples, suggesting a homogenous phenotype as defined by the fusion oncoprotein may aid in identifying germline associations. Effect estimates for variants at these loci exhibit elevated odds ratios (OR > 1.7), which is high for cancer GWAS and striking in light of the rarity of EwS in familial cancer predisposition syndromes [18]. Most EwS susceptibility loci reside near GGAA microsatellites and may disrupt local binding of EWSR1-ETS fusion oncoproteins to these microsatellites, suggesting germline-somatic interactions could be important for EwS susceptibility. As a proof-of-concept such germline-somatic interaction has been demonstrated for the chr10 EwS susceptibility gene *EGR2* [11].

Despite recent efforts to characterize the genetic architecture of EwS, thus far, no study has investigated the contribution of low-frequency variants (minor allele frequencies (MAF) < 0.05) to EwS risk. The high locus-to-case discovery ratio of previous EwS GWAS and large effect sizes of common EwS susceptibility loci led our group to revisit whether current series of EwS cases would be sufficient to detect associations between rare or low-frequency variants and EwS risk. We systematically scanned across the genome for well-imputed, low-frequency variants associated with EwS susceptibility in the largest collection of genotyped EwS cases to date (733 EwS cases and 1,346 controls) [17].

## Materials and methods

### Study populations

The study population for the current association analysis has been described previously [17]. In brief, EwS cases were obtained from five sources: a study published by Postel-Vinay et al. [19], the Institut Curie, the Childhood Cancer Survivor Study (CCSS), the Center for Cancer Research (CCR) at the National Cancer Institute (NCI), and the NCI Bone Disease and Injury Study [20]. Ancestry of these EwS cases was estimated using SNPWEIGHTS based on SNPs found to be suitable for inferring population structure [21]. EwS cases with less than 80% European ancestry were excluded resulting in a combined set of 733 EWS cases. A total of 1,346 principal-component-matched, cancer-free controls were selected from the NCI Prostate Lung Colorectal and Ovarian Cancer Screening trial [22], American Cancer Society Cancer Prevention Study II [23], and the Spanish Bladder Cancer Study [24] for the final analysis and included with controls previously used by Postel-Vinay et al [19]. Each study participant provided informed consent, and approval to conduct this research was granted by the Institution Review Board of Institut Curie, National Cancer Institute, as well as 26 participating institutions for CCSS.

### Genotyping and quality control

For the Postel-Vinay study, DNA from tumor tissue, blood, and bone marrow was isolated using proteinase K lysis followed by phenol chloroform extraction. Genomic DNA was

genotyped by 610 Quadevl arrays (Illumina). For CCSS samples, blood DNA was isolated using the Gentra PureGene Blood kit (QIAGEN) and saliva DNA was extracted using the Oragene kit (DNA Genotek). Whole genome amplification (WGA) was performed for samples without sufficient DNA. For CCSS samples, genotyping was performed at the NCI Cancer Genomics Research Laboratory (CGR) on the Infinium Human Omni5Exome array (Illumina). The remainder of NCI and Institut Curie samples were genotyped by CGR using the OmniExpress-24 v1.1 array (Illumina).

All genotyping was performed according to standard manufacturer protocols. In brief, WGA was performed on 400 ng DNA, and the amplified DNA was fragmented, precipitated, resuspended, and hybridized to the designated arrays. Single-base extension of probes using captured DNA as template was subsequently carried out with fluorophore-conjugated nucleotides. Arrays were then scanned by iScan (Illumina) and SNPs called by GenomeStudio (Illumina). Our downstream quality control included filtering out samples with abnormal heterozygosity rate, sex discordance, <95% completion rates, and unexpected relatedness (IBD > 10%).

Genotype imputation was performed in three sets: (1) the Postel-Vinay study, (2) the CCSS EwS cases and matched controls, and (3) all remaining NCI and Institut Curie samples. All samples were pre-phased using SHAPEIT [25] and imputed using IMPUTE2 [26]. The 1,000 Genomes Phase 3 was used as the reference [27] resulting in 16,367,531 SNPs. Among these SNPs, 10,216,839 were low-frequency variants with MAF < 0.05.

### PCR validation of genotypes

Imputed genotypes for the three EwS-associated low-frequency or rare variants (rs78119607, rs112837127, rs2296730) were validated by allele-specific TaqMan assay (Thermo Fisher Scientific) at CGR following standard manufacturer protocols. The 325 samples used for validation were selected based on imputed genotype, study, and amount of available DNA.

### Statistical analysis

For each variant, we report an estimate of the odds ratio (OR), 95% confidence interval (CI), and P-value ($p_{MH}$) using a Mantel-Haenszel Test where subjects are stratified by study (e.g. CCSS, Postel-Vinay, etc.), and, when stated, the genotype at linked neighboring variant(s). Because we focused on less common variants, we used a dominant model (i.e., genotype defined as presence versus absence of rare variant) and an exact, conditional test (mantelhaen. test(exact = T)) [28,29]. We used $p_{MH} < 5 \times 10^{-8}$ to define initial GWAS significance and $p_{MH} < 0.05/1684 = 1.09 \times 10^{-5}$ for conditional tests, where 1,684 is the number of SNPs with MAF < 0.05 and $R^2 > 0.004$ with one of 6 previously identified SNPs. Potential interaction between low frequency SNPs and common SNPs were examined by logistic regression models with case-control status as outcome, low frequency and common SNPs as well as an interaction term between them as predictors. All statistical tests were two-sided and performed in R v.3.6.2 [28]. We did not investigate associations with significant variants and clinical data as limited clinical data were available for the participating EwS cases.

### Results

Our analysis identified evidence for associations of three putative low frequency (MAF < 0.05) imputed variants associated with EwS risk, which we advanced to validation studies described below. The variants were located at 1q23.3, 20p11.23, and 20p11.22 (Table 1, Fig 1 and S1 Fig) and tagged by rs78119607, rs112837127, and rs2296730, respectively. The MAF among controls of European ancestry ranged from 0.001 for rs78119607 to 0.046 for rs2296730 with

**Table 1. Genome-wide significant associations (P-value < 5×10⁻⁸) for identified low-frequency and rare variants with EwS susceptibility using a dominant model stratified by study.**

| Region | Coordinate | Variant | Alleles | | Minor Allele Counts (Frequency) | | MH P-value |
|---|---|---|---|---|---|---|---|
| | | | Major | Minor | Controls N = 1,346 | EwS Cases N = 733 | |
| 1q23.3 | 163530987 | rs78119607 | G | A | 4 (0.001) | 31 (0.021) | $2.38 \times 10^{-11}$ |
| 20p11.23 | 21063508 | rs112837127 | G | A | 87 (0.032) | 9 (0.006) | $6.90 \times 10^{-9}$ |
| 20p11.22 | 21367741 | rs2296730 | A | G | 123 (0.046) | 133 (0.091) | $4.92 \times 10^{-8}$ |

minor allele effect sizes ranging from 0.18 to 16.64 (Table 2). The odds ratio for the minor A allele of rs112837127 suggested a potentially protective effect (OR = 0.18) indicating that in some instances low-frequency variation could reduce susceptibility to EwS.

To validate the imputed genotypes of the three associated low-frequency and rare variants, we first examined the imputation quality score (S1 Table) and distribution of alleles (S2 Table) across three studies populations, and we did not observe significant heterogeneity among the study populations. To further confirm the findings, an allele-specific TaqMan assay was designed for the three variants and carried out in a subset of 325 samples from the EwS GWAS with available remaining DNA. As shown in S2 Fig, we were able to replicate the imputed genotypes for rs112837127 and rs2296730 with 98.46% and 100% concordance rate. The imputed genotype for rs78119607 did not replicate as no minor alleles were called by the Taq-Man assay, suggesting poor imputation of this variant using the 1000 Genomes Project reference set despite imputation scores of over 0.43 (S1 Table).

The two validated low frequency variants, rs112837127 and rs2296730, associated with EwS on chromosome 20 are in proximity to two previously identified EwS common susceptibility variants, rs6106336 and rs6047482. The identified low-frequency variants were tested for linkage disequilibrium (LD) with the common variants in 1000 Genomes Project European populations using the LDmatrix tool in LDlink (Fig 2) [30,31]. rs112837127 did not display evidence for LD with either the nearby common variant ($R^2_{EUR}$ rs6106336 = 0.005, $R^2_{EUR}$ rs6047482 = 0.023) or the other low-frequency variant ($R^2_{EUR}$ rs2296730 = 0.003). However, rs2296730 displayed evidence for moderate levels of LD with the common rs6106336 variant ($R^2_{EUR}$ = 0.311), but not the common rs6047482 variant ($R^2_{EUR}$ = 0.006). Estimates of D′, a measure of allelic transmission, suggest the two associated low-frequency variants (rs112837127 and rs2296730) are transmitted on haplotypes of the common rs6106336 variant (S3 Fig), with the minor A allele of rs112837127 being transmitted with the major T allele of rs6106336 ($D'_{EUR}$ = 1.0) and the minor G allele of rs2296730 being transmitted with the minor G allele of the rs6106336 ($D'_{EUR}$ = 0.772).

To further test if the two low-frequency variants tagged independent EwS association signals, odds ratios and P-values for the association with EwS were calculated with and without conditioning on the neighboring common variants. Conditional analyses indicated that rs112837127 was statistically associated with EwS (OR = 0.20, 95%CI = 0.09–0.40, P-value = $5.84 \times 10^{-8}$; Table 2) independent from neighboring common variants. As in the $R^2$ analyses, the low-frequency rs22966730 variant demonstrated evidence for a correlation with the common rs6106336 variant as observed in the attenuated odds ratio estimate and increase in p-value in the conditional analysis (OR = 1.61, 95%CI = 1.16–2.24, P-value = $3.50 \times 10^{-3}$; Table 2). Finally, we examined potential interaction between rs2296730 and rs6106336 (p = 0.568), rs2296730 and rs6047482 (p = 0.319), as well as rs6106336 and rs112837127 (p = 0.538) and found no significant evidence for SNP-SNP interactions.

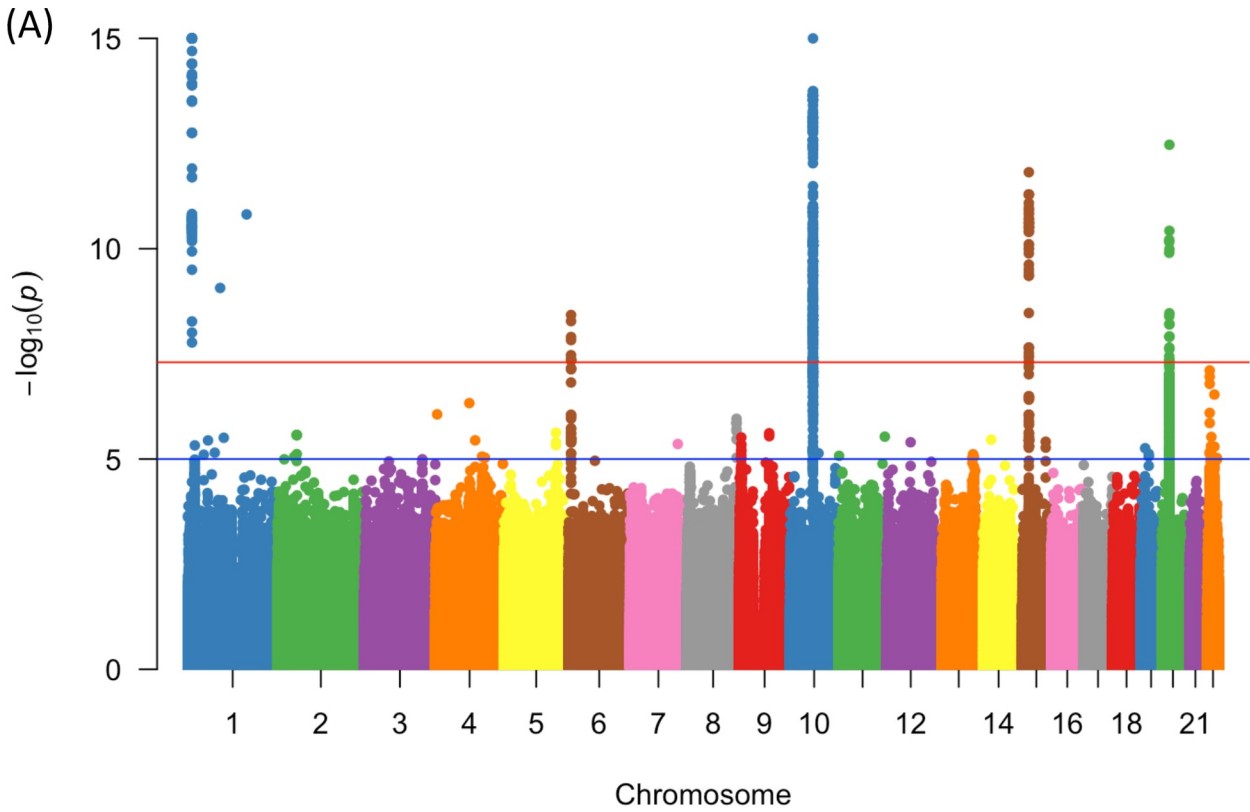

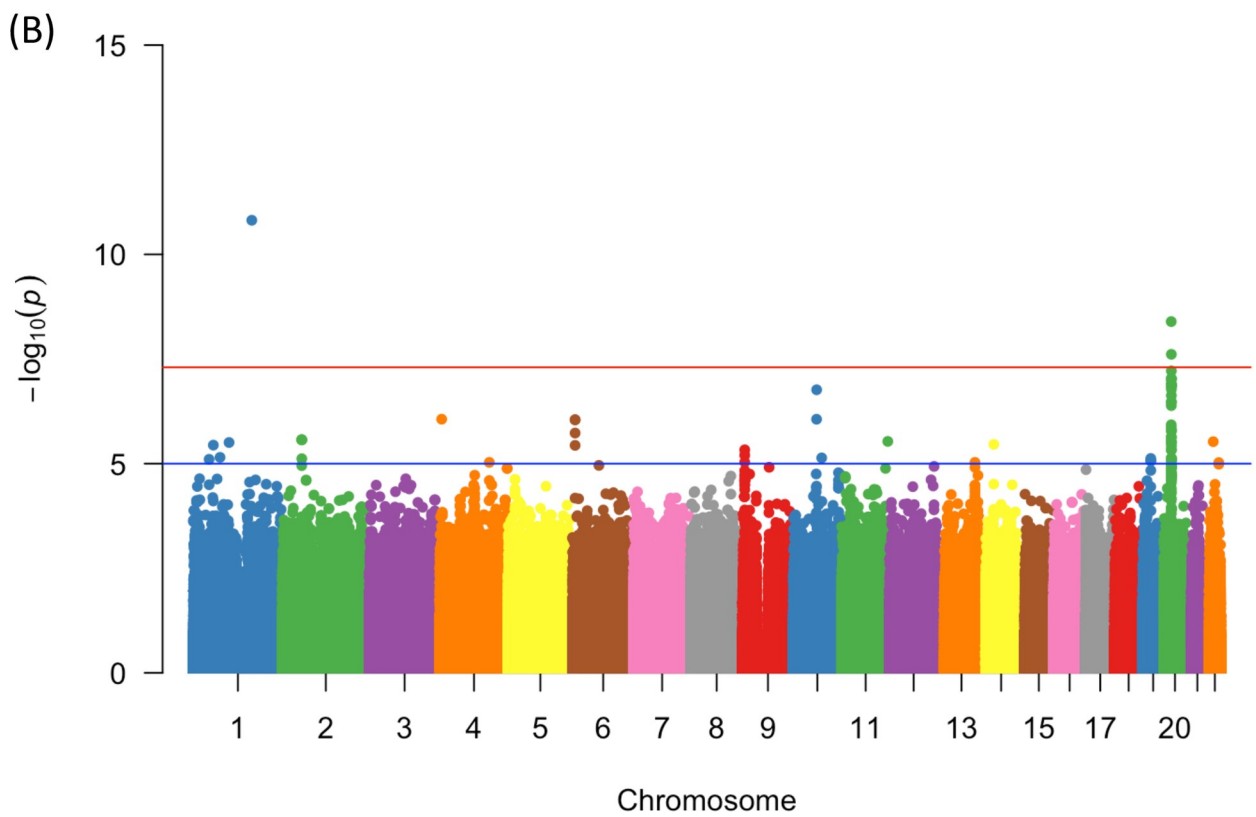

**Fig 1.** Manhattan plots of analyses for all variants (**A**) and low-frequency and rare variants (MAF < 0.05) (**B**). Plotted p-values are for allelic tests by chromosome.

## Discussion

We report an analysis of well-imputed low-frequency variants based on common genotyped variants in a large EwS case series to investigate the contribution of low-frequency variants to the underlying genetic architecture of EwS susceptibility. We found evidence for associations of two low-frequency variants (rs112837127 and rs22966730) with EwS risk, and one of the variants, rs112837127, demonstrated an association independent of a nearby common germline susceptibility variant. Our findings suggest that in addition to common germline susceptibility variants, low-frequency variants are important for genetic susceptibility to EwS. Germline variants associated with lower cancer risk are less commonly reported, but not unheard of. Previously, three SNPs located near base excision repair genes were found to be negatively associated with Wilms tumor risk [32]. SNPs in the vitamin D receptor gene have also been linked to decreased risk in prostate cancer in African American men [33] and rs1866074 near the thymine DNA glycosylase gene were reported to be correlated with lower colorectal cancer risk [34]. The minor allele of rs112837127 is most prevalent in British and Finnish populations where the allele frequency could be > 5% while no African or east Asian population carries this allele [35]. This SNP is located in a long terminal repeat region 2.7 Kb upstream of a non-coding RNA, LINC00237, which has been found to drive self-renewal of tumor initiating cells by binding and promoting stability of β-catenin [36]. Interestingly, the activation of Wnt/β-catenin pathway has been shown to antagonize transcription activities of EWS/ETS fusion gene in Ewing sarcoma cells [37]. Whether the minor allele of rs112837127 tags a haplotype with modified LINC00237 expression remains to be investigated.

As EwS is a rare sarcoma of young people, it is not unexpected that low-frequency variation contributes to EwS susceptibility. Although EwS may be an exceptional case of a rare, well-defined malignancy with high associated odds ratios, our study suggests that efforts to examine low-frequency and rare germline associations in existing samples of rare cancer sets could be fruitful, even despite limited sample sizes. Additionally, our study provides an example in which common germline susceptibility loci discovered by GWAS may harbor synthetic associations with rare and low-frequency variants [28]. These synthetic associations may be of particular importance for EwS susceptibility as it is plausible common, low-frequency and rare variation at GGAA microsatellites may interact to impact binding of EWSR1-FLI1 fusion oncoproteins and alter regulation of downstream genes in core EwS regulatory pathways. In the case of EwS, common variant associations may highlight important EwS germline susceptibility regions where low-frequency and rare variation have important roles altering EwS risk.

**Table 2. Estimated odds ratio (OR) for EwS rare variants adjusting for different model covariates.**

| Rare SNP | Common SNP | Wald method (unadjusted) | | Mantel-Haenszel (study) | | Mantel-Haenszel (study and variant[1]) | |
|---|---|---|---|---|---|---|---|
| | | OR (95% CI) | Fisher's P-value | OR (95% CI) | P-value | OR (95% CI) | P-value |
| rs112837127 | rs6106336 | 0.19 (0.10 to 0.39) | $1.64\times10^{-8}$ | 0.18 (0.08 to 0.37) | $6.90\times10^{-9}$ | 0.20 (0.09 to 0.40) | $5.84\times10^{-8}$ |
| rs2296730 | rs6106336, rs6047482 | 2.04 (1.58 to 2.69) | $9.78\times10^{-9}$ | 2.11 (1.60 to 2.77) | $4.92\times10^{-8}$ | 1.61 (1.16 to 2.24) | $3.50\times10^{-3}$ |

Models use a dominant allele coding for minor alleles with each individual as the analysis unit.

[1]Adjustment for contributing study and nearby common SNP(s).

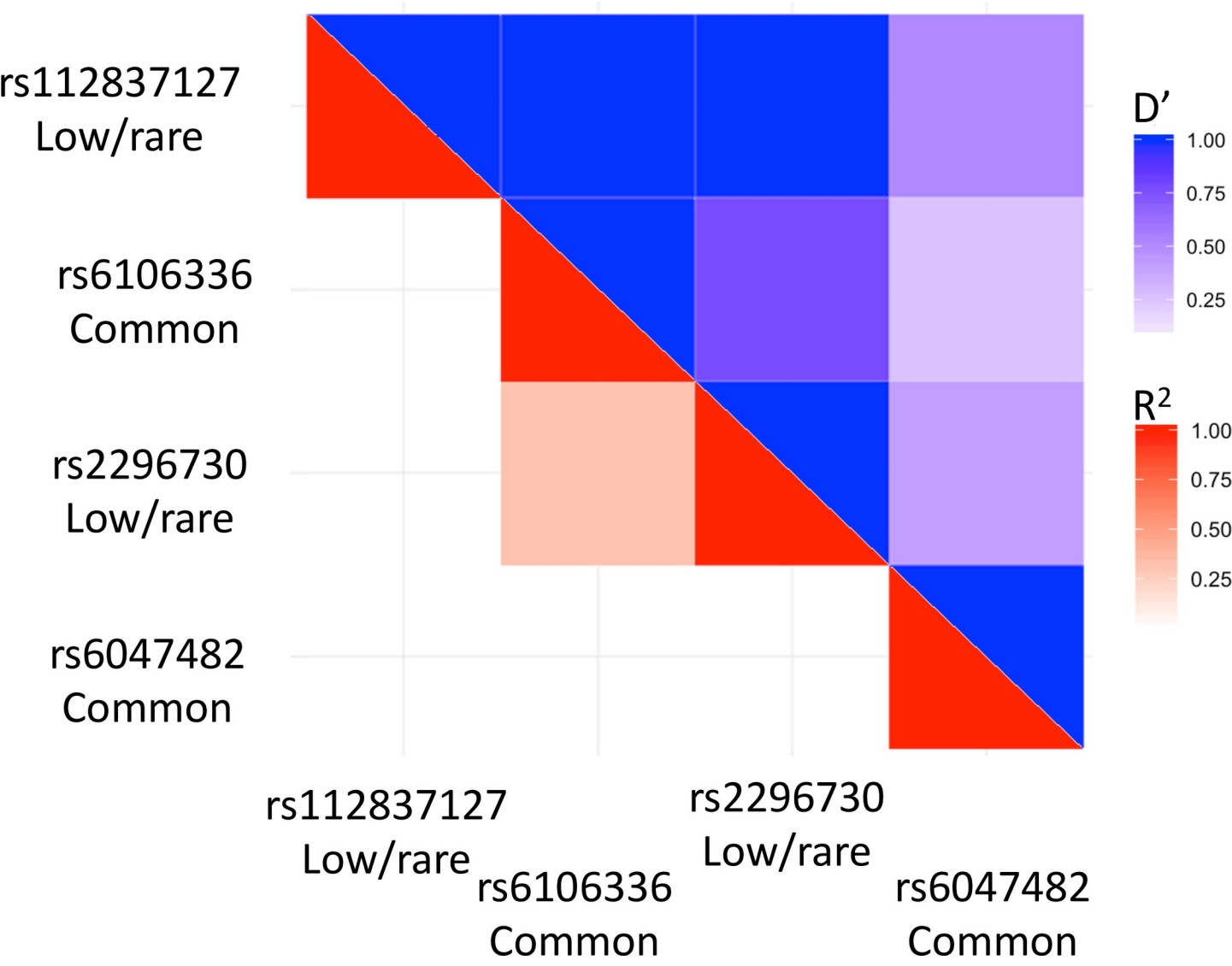

**Fig 2. Patterns of Linkage Disequilibrium (LD) for rare, low-frequency and common variants associated with EwS at the chromosome 20p11.22–23 susceptibility locus.** $R^2$ values are in shades of red while D' values are in shades of blue, with darker values indicating higher degree of LD. All LD measures were estimated in LDlink using 1,000 Genomes Project European populations as the reference panel.

A limitation of our study is the lack of validation in an independent cohort as well as a lack of regional EwS sequencing of the relevant region to identify potential causal variants which can be functionally examined through *in vitro* experiments. Another limitation is the absence of clinical and demographic data which limited our ability to describe possible associations with the variants identified. As EwS is a rare tumor, few large case series exist for genomic investigation. Larger study populations will be essential for further confirmation of this new association. As future germline association studies investigate the genetic architecture of EwS, improved efforts to systematically interrogate low-frequency variant associations through a variety of sequencing and statistical methods are essential for accelerating understanding of the underlying genetic architecture of EwS susceptibility.

## Supporting information

**S1 Fig. LDassoc regional association plots for identified rare and low-frequency variant associations with EwS susceptibility.** Plots are for rs78119607 (**A**), rs112837127 (**B**), and rs2296730 (**C**).
(DOCX)

**S2 Fig. Validation results of EwS associated rare and low-frequency variants by TaqMan assays.**
(DOCX)

**S3 Fig.** Linkage disequilibrium between the common variant rs6106336 and the two identified low frequency variants (A) rs112837127 and (B) rs2296730 using LDpair and all European 1,000 Genomes populations as a reference.
(DOCX)

**S1 Table. Imputation quality scores for each associated low-frequency or rare variant by EwS imputation set.**
(DOCX)

**S2 Table. Distribution of alleles across three EwS study populations.**
(DOCX)

## Acknowledgments

We thank the following clinicians for providing samples used in this study: C. Alenda, F. Almazán, D. Ansoborlo, L. Aymerich, L. Benboukbher, C. Beléndez, C. Berger, C. Bergeron, P. Biron, J. Y. Blay, E. Bompas, H. Bonnefoi, P. Boutard, B. Bui-Nguyen, D. Chauveaux, C. Calvo, A. Carboné, C. Clement, T. Contra, N. Corradini, A. S. Defachelles, V. Gandemer-Delignieres, A. Deville, A. Echevarria, J. Fayette, M. Fraga, D. Frappaz, J. L. Fuster, P. García-Miguel, J. C. Gentet, P. Kerbrat, V. Laithier, V. Laurence, P. Leblond, O. Lejars, R. López-Almaraz, B. López-Ibor, P. Lutz, J. F. Mallet, L. Mansuy, P. Marec Bérard, G. Margueritte, A. Marie Cardine, C. Melero, L. Mignot, F. Millot, O. Minckes, G. Margueritte, C. Mata, M. E. Mateos, M. Melo, C. Moscardó, M. Munzer, B. Narciso, A. Navajas, D. Orbach, C. Oudot, H. Pacquement, C. Paillard, Y. Perel, T. Philip, C. Piguet, M. I. Pintor, D. Plantaz, E. Plouvier, S. Ramirez-Del-Villar, I. Ray-Coquard, Y. Reguerre, M. Rios, P. Rohrlich, H. Rubie, A. Sastre, G. Schleiermacher, C. Schmitt, P. Schneider, L. Sierrasesumaga, C. Soler, N. Sirvent, S. Taque, E. Thebaud, A. Thyss, R. Tichit, J. J. Uriz, J. P. Vannier, F. Watelle-Pichon.

## Author Contributions

**Conceptualization:** Mitchell J. Machiela.

**Data curation:** Thomas G. P. Grünewald, Didier Surdez, Stephanie Reynaud, Olivier Mirabeau, Eric Karlins, Rebeca Alba Rubio, Sakina Zaidi, Sandrine Grossetête-Lalami, Stelly Ballet, Eve Lapouble, Valérie Laurence, Jean Michon, Gaelle Pierron, Heinrich Kovar, Udo Kontny, Anna González-Neira, Javier Alonso, Ana Patino-Garcia, Nadège Corradini, Perrine Marec Bérard, Jeremy Miller, Neal D. Freedman, Nathaniel Rothman, Brian D. Carter, Casey L. Dagnall, Laurie Burdett, Kristine Jones, Michelle Manning, Kathleen Wyatt, Weiyin Zhou, Meredith Yeager, David G. Cox, Robert N. Hoover, Javed Khan, Gregory T. Armstrong, Wendy M. Leisenring, Smita Bhatia, Leslie L. Robison, Andreas E. Kulozik, Jennifer Kriebel, Thomas Meitinger, Markus Metzler, Manuela Krumbholz, Wolfgang Hartmann, Konstantin Strauch, Thomas Kirchner, Uta Dirksen, Lisa Mirabello, Margaret

A. Tucker, Franck Tirode, Lindsay M. Morton, Stephen J. Chanock, Olivier Delattre, Mitchell J. Machiela.

**Formal analysis:** Shu-Hong Lin, Jeremy Miller, Mitchell J. Machiela.

**Investigation:** Shu-Hong Lin.

**Methodology:** Joshua N. Sampson, Jeremy Miller.

**Supervision:** Mitchell J. Machiela.

**Visualization:** Shu-Hong Lin.

**Writing – original draft:** Shu-Hong Lin.

**Writing – review & editing:** Shu-Hong Lin, Joshua N. Sampson, Thomas G. P. Grünewald, Didier Surdez, Stephanie Reynaud, Olivier Mirabeau, Eric Karlins, Rebeca Alba Rubio, Sakina Zaidi, Sandrine Grossetête-Lalami, Stelly Ballet, Eve Lapouble, Valérie Laurence, Jean Michon, Gaelle Pierron, Heinrich Kovar, Udo Kontny, Anna González-Neira, Javier Alonso, Ana Patino-Garcia, Nadège Corradini, Perrine Marec Bérard, Jeremy Miller, Neal D. Freedman, Nathaniel Rothman, Brian D. Carter, Casey L. Dagnall, Laurie Burdett, Kristine Jones, Michelle Manning, Kathleen Wyatt, Weiyin Zhou, Meredith Yeager, David G. Cox, Robert N. Hoover, Javed Khan, Gregory T. Armstrong, Wendy M. Leisenring, Smita Bhatia, Leslie L. Robison, Andreas E. Kulozik, Jennifer Kriebel, Thomas Meitinger, Markus Metzler, Manuela Krumbholz, Wolfgang Hartmann, Konstantin Strauch, Thomas Kirchner, Uta Dirksen, Lisa Mirabello, Margaret A. Tucker, Franck Tirode, Lindsay M. Morton, Stephen J. Chanock, Olivier Delattre, Mitchell J. Machiela.

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
