## [Decision Letter · Decision Letter 0]

15 May 2020

PONE-D-20-05330

Low-frequency variation near common germline susceptibility loci are associated with risk of Ewing sarcoma

PLOS ONE

Dear Dr. Machiela,

Thank you for submitting your manuscript to PLOS ONE. After careful consideration, we feel that it has merit but does not fully meet PLOS ONE’s publication criteria as it currently stands. Therefore, we invite you to submit a revised version of the manuscript that addresses the points raised during the review process.

We would appreciate receiving your revised manuscript by Jun 29 2020 11:59PM. To enhance the reproducibility of your results, we recommend that if applicable you deposit your laboratory protocols in protocols.io, where a protocol can be assigned its own identifier (DOI) such that it can be cited independently in the future. For instructions see: http://journals.plos.org/plosone/s/submission-guidelines#loc-laboratory-protocols

We look forward to receiving your revised manuscript.

Kind regards,

Yanhong Liu

Academic Editor

PLOS ONE

Journal Requirements:

"Each study participant provided informed consent and each participating study was approved by the Institutional Review Boards of their respective study center."

b. Once you have amended this statement in the Methods section of the manuscript, please add the same text to the “Ethics Statement” field of the submission form (via “Edit Submission”).

'The authors have declared that no competing interests exist.'  

We note that one or more of the authors are employed by a commercial company: Leidos Biomedical Research Inc. and Information Management Services, Inc.

Reviewers' comments:

Reviewer's Responses to Questions

**Comments to the Author**

1. Is the manuscript technically sound, and do the data support the conclusions?

Reviewer #1: Yes

Reviewer #2: No

Reviewer #3: Yes

2. Has the statistical analysis been performed appropriately and rigorously? 

Reviewer #1: Yes

Reviewer #2: No

Reviewer #3: Yes

3. Have the authors made all data underlying the findings in their manuscript fully available?

Reviewer #1: Yes

Reviewer #2: Yes

Reviewer #3: Yes

4. Is the manuscript presented in an intelligible fashion and written in standard English?

Reviewer #1: Yes

Reviewer #2: Yes

Reviewer #3: Yes

5. Review Comments to the Author

Reviewer #1: This paper presents genome-wide association studies of Ewing sarcoma. The authors identified two low-frequency variants, rs112837127 and rs2296730, on chromosome 20 that were associated with EwS risk (OR = 0.186 and 2.038, respectively; P-value < 5×10 -8). The work is meaningful to the cancer therapy, but there are some modifications.

1) Can the authors provide clinical indexes for the studies?

2) A statistical test should be made between genotypes and clinical indexes.

3) The manuscript lacks discussion section. Results section and discussion section should be separated.

4) A cox risk regression model should be made based on current result.

Reviewer #2: The author utilized imputed genotype data from several studies and identified two rare variants that may be associated with EwS. However, the analysis is not solid enough to support the conclusion and additional work is required to verify the findings.

1. The author used data from five sources and combine them together for the imputation and association test. However, the author didn’t mention how he/she combine the data together and how did he/she deal with the batch effects caused by different source of data.

2. The type I error is not estimated in the paper, which is necessary, especially for merging data from different sources.

3. It is also necessary to clarity the cutoff value that the author used to control the imputation quality.

Reviewer #3: Lin and colleagues present a GWAS in Ewing sarcoma identifying two rare variants within previously identified risk loci that modulate risk for this disease. The manuscript is well written and all methodologies used are appropriate, and these results represent an important contribution to our understanding of genetic susceptibility to this rare and deadly cancer.

I have a few minor comments and questions

1. Please note what program was used for statistical analysis

2. Please include a supplementary table showing basic demographic information for individuals included in this analysis, particularly the sex and age distributions as controls were obtained from separate source.

3. It would be helpful to include the total number of SNPs in the dataset after imputation.

4. While it is likely difficult to assess due to the small sample size, did you identify any variation in the allele frequencies of these rare SNPs by age? For example, were those diagnosed at younger ages more likely to be carriers of these SNPs?

5. Did you evaluate whether there was any interaction between these rare SNPs and previously identified common SNPs at these loci?

6. PLOS authors have the option to publish the peer review history of their article (what does this mean?). If published, this will include your full peer review and any attached files.

Reviewer #1: No

Reviewer #2: No

Reviewer #3: No

---

## [Author Response · Author response to Decision Letter 0]

29 Jun 2020

Reviewer #1:

This paper presents genome-wide association studies of Ewing sarcoma. The authors identified two low-frequency variants, rs112837127 and rs2296730, on chromosome 20 that were associated with EwS risk (OR = 0.186 and 2.038, respectively; P-value < 5×10 -8). The work is meaningful to the cancer therapy, but there are some modifications.

We thank Reviewer #1 for their time reviewing our manuscript. We agree that our manuscript is meaningful to Ewing sarcoma research and have addressed their comments with the below response and modifications.

1) Can the authors provide clinical indexes for the studies?

We agree with Reviewer #1 that it would be of interest to investigate clinical indices associated with Ewing sarcoma with these two reported variants (rs112837127 and rs2296730) to better understand potential clinical relationships. However, as Ewing sarcoma is a rare malignancy we had to collect cases from across many different recruitment centers and as such have very heterogeneous and limited clinical information for these participating Ewing sarcoma cases. Any effort to collect additional clinical information would require substantial new effort and likely would still result in high amounts of missing clinical information. Even with complete clinical information, association analyses would likely not be very fruitful as our case series of this rare tumor is limited in size and investigating clinical associations with rare variants can be challenging due to limited statistical power to detect associations.

To address this comment we have added the following text to the Methods section:

“We did not investigate associations with significant variants and clinical data as limited clinical data was available for the participating EwS cases.”

As well as the following text to the Discussion section:

“Another limitation is the absence of clinical and demographic data which limited our ability to describe possible associations with the variants identified.”

2) A statistical test should be made between genotypes and clinical indexes.

The main focus of our investigation is on susceptibility of EwS and as such we did not perform analyses on clinical indices. In general, genotype-based tests for low-frequency variants (<5% minor allele frequency) are challenging to perform as Ewing sarcoma is a rare tumor for which to amass a large sample size and a small fraction of individuals are expected to be homozygous for the rare allele (<0.25%). Please see above response for question 1 for additional details on the challenge of collecting clinical data. We agree that future such studies that address the relationship between germline susceptibility variants and clinical indices are needed, but at this time resources do not exist to establish such a study and perform these analyses.

To address this comment, we have added the following text to the Methods section:

“We did not investigate associations with significant variants and clinical data as limited clinical data was available for the participating EwS cases.”

As well as the following text to the Discussion section:

“Another limitation is the absence of clinical and demographic data which limited our ability to describe possible associations with the variants identified.”

3) The manuscript lacks discussion section. Results section and discussion section should be separated.

We thank Reviewer #1’s suggestion and have accordingly revised the manuscript to include separate Results and Discussion sections in our manuscript.

4) A cox risk regression model should be made based on current result.

We are unclear what Reviewer #1 is suggesting as this is a case-control study and we have no available time-to-event data for which to run a Cox model.

Reviewer #2:

The author utilized imputed genotype data from several studies and identified two rare variants that may be associated with EwS. However, the analysis is not solid enough to support the conclusion and additional work is required to verify the findings.

We thank Reviewer #2 for their time reviewing our manuscript and are appreciative of their thoughtful comments. Please see the below text for our detailed responses to their comments.

1. The author used data from five sources and combine them together for the imputation and association test. However, the author didn’t mention how he/she combine the data together and how did he/she deal with the batch effects caused by different source of data.

We agree with Reviewer #2 that it is of paramount importance to account for potential study or batch effects in our investigation, particularly in the analysis of our study. To ensure uniform imputation of our study, we only used high-quality genotypes as input for the imputation process and used the same 1000 Genomes Project reference panel for each study. We then employed a Mantel-Haenzel test which adjusts for difference among studies by estimating effects within each study strata and then combined the stratified results together into a combined estimate. This is a robust approach to account for potential batch differences by study. We have described the approach in the text of the Methods section:

“For each variant, we report an estimate of the odds ratio (OR), 95% confidence interval (CI), and P-value (pMH) using a Mantel-Haenszel Test where subjects are stratified by study (e.g. CCSS, Postel-Vinay, etc.)”

2. The type I error is not estimated in the paper, which is necessary, especially for merging data from different sources.

We agree with Reviewer #2 that adjusting for multiple comparisons in our analysis is important. As such, we have used a conservative Bonferroni-based cutoff of genome-wide significance defined as a p-value less than 5×10-8. This is an industry-based standard p-value threshold estimated on genome-wide LD patterns to ensure low false positive rates from genome-wide association studies. The merging of data from different sources in our analysis does not impact the type I error as the same number of variants are still being investigated, it simply boosts our power to better investigate whether each variant is associated as information from more cases and controls is available. Furthermore, the resulting independent associations of low-frequency variants rs112837127 and rs2296730 near the known Ewing sarcoma chromosome 20 susceptibility locus adds to the evidence that this region is important for Ewing sarcoma susceptibility and suggests that common and low-frequency germline variation interact in this region to impact Ewing sarcoma risk.

We have described the p-value significance threshold in our manuscript as follows:

“We used pMH < 5 × 10-8 to define initial GWAS significance and pMH < 0.05/1684=1.09×10-5 for conditional tests, where 1,684 is the number of SNPs with MAF < 0.05 and R2 > 0.004 with one of 6 previously identified SNPs.”

3. It is also necessary to clarity the cutoff value that the author used to control the imputation quality.

We did not filter SNPs by imputation quality scores, but we did examine the quality score for our candidate SNPs as shown in table S1 and described in the following text in results:

“To validate the imputed genotypes of the three associated low-frequency and rare variants, we first examined the imputation quality score (Supplementary Table 1) and distribution of alleles (Supplementary Table 2) across three studies populations, and we did not observe significant heterogeneity among the study populations.”

Reviewer #3:

Lin and colleagues present a GWAS in Ewing sarcoma identifying two rare variants within previously identified risk loci that modulate risk for this disease. The manuscript is well written and all methodologies used are appropriate, and these results represent an important contribution to our understanding of genetic susceptibility to this rare and deadly cancer.

We thank Reviewer #3 for their careful review of our manuscript and are pleased they found our manuscript to represent an important contribution to our understanding of Ewing sarcoma genetic susceptibility.

I have a few minor comments and questions

1. Please note what program was used for statistical analysis

The program we used is described as follows in the revised methods section:

“All statistical tests were two-sided and performed in R v.3.6.2 (28)”

2. Please include a supplementary table showing basic demographic information for individuals included in this analysis, particularly the sex and age distributions as controls were obtained from separate source.

We understand the interest in knowing additional demographic information from our study participants. Unfortunately, as Ewing sarcoma is a rare tumor it required the recruitment of participants across many years from multiple study centers that provided varying amounts of information on cases. As such, considerable effort would be needed to individually recover information from each participant that likely would not be fruitful in producing a complete dataset. As the main focus of our analysis is on genetic susceptibility to Ewing sarcoma, we feel the omission of this information does not significantly impact the results of our association analysis and goes beyond the original intended scope of our research question. Please also refer to our response for comment 1 from Reviewer #1.

3. It would be helpful to include the total number of SNPs in the dataset after imputation.

We appreciate Reviewer #3’s suggestion and have added the total number of SNPs after imputation to our manuscript. The following text in the Methods section details the SNPs included in our analysis:

“Genotype imputation was performed in three sets: (1) the Postel-Vinay study, (2) the CCSS EwS cases and matched controls, and (3) all remaining NCI and Institut Curie samples. All samples were pre-phased using SHAPEIT (25) and imputed using IMPUTE2 (26). The 1,000 Genomes Phase 3 was used as the reference (27) resulting in 16,367,531 SNPs. Among these SNPs, 10,216,839 were low-frequency variants with MAF < 0.05.”

4. While it is likely difficult to assess due to the small sample size, did you identify any variation in the allele frequencies of these rare SNPs by age? For example, were those diagnosed at younger ages more likely to be carriers of these SNPs?

As mentioned above, we did not have age at diagnosis for most cases so were unable to run this analysis. We agree with Reviewer #3 that even if we had this data this question would likely be difficult to assess due to the small sample size.

5. Did you evaluate whether there was any interaction between these rare SNPs and previously identified common SNPs at these loci?

We thank Reviewer #3’s insight and performed logistic regression using Ewing sarcoma diagnosis as outcome, and the SNPs of interest as predictors to test potential interaction between rs2296730 and rs6106336 (p=0.568), rs2296730 and rs6047482 (p=0.319), as well as rs6106336 and rs112837127 (p = 0.538). None of these combinations reached statistical significance suggesting no interaction between these SNPs and surrounding common SNPs associated with Ewing sarcoma. These results as well as methods has been described in the revised manuscript:

“Potential interaction between low frequency SNPs and common SNPs were examined by logistic regression models with case-control status as outcome, low frequency and common SNPs as well as an interaction term between them as predictors.” -- Methods

“Finally, we examined potential interaction between rs2296730 and rs6106336 (p=0.568), rs2296730 and rs6047482 (p=0.319), as well as rs6106336 and rs112837127 (p = 0.538) and found no significant evidence for SNP-SNP interactions.” -- Results

---

## [Decision Letter · Decision Letter 1]

4 Aug 2020

Low-frequency variation near common germline susceptibility loci are associated with risk of Ewing sarcoma

PONE-D-20-05330R1

Dear Dr. Machiela,

We’re pleased to inform you that your manuscript has been judged scientifically suitable for publication and will be formally accepted for publication once it meets all outstanding technical requirements.

Kind regards,

Yanhong Liu

Academic Editor

PLOS ONE

Additional Editor Comments (optional):

Reviewers' comments:

Reviewer's Responses to Questions

**Comments to the Author**

1. If the authors have adequately addressed your comments raised in a previous round of review and you feel that this manuscript is now acceptable for publication, you may indicate that here to bypass the “Comments to the Author” section, enter your conflict of interest statement in the “Confidential to Editor” section, and submit your "Accept" recommendation.

Reviewer #1: All comments have been addressed

Reviewer #3: All comments have been addressed

2. Is the manuscript technically sound, and do the data support the conclusions?

Reviewer #1: Yes

Reviewer #3: Yes

3. Has the statistical analysis been performed appropriately and rigorously? 

Reviewer #1: Yes

Reviewer #3: Yes

4. Have the authors made all data underlying the findings in their manuscript fully available?

Reviewer #1: Yes

Reviewer #3: Yes

5. Is the manuscript presented in an intelligible fashion and written in standard English?

Reviewer #1: Yes

Reviewer #3: Yes

6. Review Comments to the Author

Reviewer #1: All the comments have been addressed. The submission has been greatly improved and is worthy of publication.

Reviewer #3: (No Response)

7. PLOS authors have the option to publish the peer review history of their article (what does this mean?). If published, this will include your full peer review and any attached files.

Reviewer #1: No

Reviewer #3: No

---

## [Editor Report · Acceptance letter]

25 Aug 2020

PONE-D-20-05330R1 

Low-frequency variation near common germline susceptibility loci are associated with risk of Ewing sarcoma 

Dear Dr. Machiela:

I'm pleased to inform you that your manuscript has been deemed suitable for publication in PLOS ONE. Congratulations! Your manuscript is now with our production department. 

Kind regards, 

on behalf of

Dr. Yanhong Liu 

Academic Editor

PLOS ONE